Manuscript prepared for Atmos. Meas. Tech. with version 5.0 of the  $L^{A}T_{E}X$  class copernicus.cls. Date: 27 April 2020

# Effects of clouds on the UV Absorbing Aerosol Index from TROPOMI

Maurits L. Kooreman<sup>1</sup>, Piet Stammes<sup>1</sup>, Victor Trees<sup>1</sup>, Maarten Sneep<sup>1</sup>, L. Gijsbert Tilstra<sup>1</sup>, Martin de Graaf<sup>1</sup>, Deborah C. Stein Zweers<sup>1</sup>, Ping Wang<sup>1</sup>, Olaf N. E. Tuinder<sup>1</sup>, and J. Pepijn Veefkind<sup>1</sup>

<sup>1</sup>Royal Netherlands Meteorological Institute, De Bilt, The Netherlands *Correspondence to:* M. L. Kooreman (kooreman@knmi.nl)

**Abstract.** The ultraviolet (UV) Absorbing Aerosol Index (AAI) is widely used as an indicator for the presence of absorbing aerosols in the atmosphere. Here we consider the TROPOMI AAI based on the 340/380 nm wavelength pair. We investigate the effects of clouds on the AAI observed at small and large scales. The large scale effects are studied using an aggregate of TROPOMI

- measurements over an area mostly devoid of absorbing aerosols (Pacific Ocean). The study reveals that several structural features can be distinguished in the AAI, such as the cloud bow, viewing zenith angle dependence, sunglint, and a previously unexplained increase in AAI values at extreme viewing and solar geometries. We explain these features in terms of the Bidirectional Reflectance Distribution Function (BRDF) of the scene in combination with the different ratio of diffuse and
- direct illumination of the surface at 340 and 380 nm. To reduce the dependency on the BRDF and homogenize the AAI distribution across the orbit, we present three different AAI retrieval models: the traditional Lambertian Scene Model (LSM), a Lambertian Cloud Model (LCM), and a Scattering Cloud Model (SCM). We perform a model study to assess the propagation of errors in auxiliary databases used in the cloud models. The three models are then applied to the same low-aerosol
- region. Results show that using the LCM and SCM gives on average a higher AAI than the LSM. Additionally, a more homogeneous distribution is retrieved across the orbit. At the small scale, related to the high spatial resolution of TROPOMI, strong local increases and decreases in AAI are observed in the presence of clouds. This effect was not observed in previous instruments with larger ground pixels such as GOME-2 and OMI. More research is needed to explain these small scale effects.

1

## 1 Introduction

Aerosols are small liquid or solid particles suspended in the air. Aerosols have a direct effect on climate, because they absorb and scatter solar and terrestrial radiation. In terms of radiative properties, two types of aerosols can be distinguished: absorbing and scattering aerosols. Absorbing aerosols,

- such as smoke from biomass burning, desert dust, volcanic ash, and anthropogenically produced soot, absorb radiation and have a warming effect on the climate. Scattering aerosols, like sulfate particles and clouds, scatter solar light and usually have a cooling effect on the climate. Aerosols also act as condensation nuclei in the process of cloud formation, potentially altering the optical properties of these clouds.
- The ultraviolet Absorbing Aerosol Index (AAI) indicates the presence of absorption in the atmosphere, attributed by aerosols. It separates the spectral contrast at two ultraviolet (UV) wavelengths caused by aerosol absorption from that of molecular Rayleigh scattering, surface reflection, and absorption by trace gases (Torres et al., 1998; de Graaf, 2002). Ideally, the AAI is zero if there are no absorbing or scattering aerosols present in the scene.
- Originally, methods of observing aerosols from space relied on measurements in the visible and infrared regions of the spectrum. In these spectral regions, Rayleigh scattering is less important and inversion calculations are relatively simple. However, developments in radiative transfer calculations resulted in the possibility to account for the multiple scattering occurring in the UV spectral region which, in turn, allowed for novel techniques of measuring aerosols. The use of UV radiation for
- global detection of aerosols has advantages, because in this spectral region most surfaces are dark, resulting in high contrast with atmospheric effects and a lower sensitivity to aerosol near the surface. The AAI as an indicator of absorbing aerosols has a strong heritage with retrievals from TOMS (Herman et al., 1997), GOME(-1) (de Graaf et al., 2005), SCIAMACHY (de Graaf and Stammes, 2005; Tilstra et al., 2012) and GOME-2 (de Graaf et al., 2017). The AAI is traditionally defined as
- the positive values of the reflectance residue between an absorbing aerosol loaded atmosphere and a clear atmosphere. Negative values are associated with an atmosphere that contains more scattering particles than a clear atmosphere (Penning De Vries et al., 2009).

Effects of clouds on the AAI were studied earlier using GOME-2 (Penning de Vries and Wagner, 2011) and OMI (Torres et al., 2018; Jethva et al., 2018) data. It was found that using the independent

pixel approximation instead of a Lambertian scene albedo improves the neutral value of the AAI for scenes with broken clouds.

Besides the monitoring of large aerosol events with TROPOMI, such as the Amazonian wildfires of 2019 and the Australian bushfires of 2019/2020, an important application of the AAI is to preselect scenes for the Aerosol Layer Height retrieval of TROPOMI (Sanders et al., 2015; Nanda et al.,

2019). Only for scenes with a positive AAI value, the ALH retrieval is performed. Since the AAI is not so much a measure of aerosols but rather a measure of the UV reflectance residue, this paper discusses the current features in the AAI product, which are mainly caused by clouds.

In Sect. 2 we discuss structural features in the AAI at small and large scales. Local features such as 3D-effects and shadows of clouds are discussed in Sect. 2.2. The large scale features, such as sunglint, cloud bow, and elevated AAI values near the orbit edges are discussed in Sect. 2.3. Section 3 discusses in-depth the theory behind the observed features and explains the different retrieval approaches discussed in Sect. 3.4, introducing two additional AAI retrieval models to represent clouds. In Sect. 4 we apply the different retrieval models to a selection of TROPOMI orbits. The results are discussed in Sect. 5. Section 6 concludes the work with a summary, a recommendation to users, and prospects of future AAI retrieval improvements.

#### 2 Cloud features observed in the TROPOMI AAI

#### 2.1 Data description

The TROPOMI instrument is a push-broom spectrometer on-board the dedicated Sentinel 5 Precursor satellite maintaining a polar orbit with an ascending node equator-crossing local time of 13:30.

- The TROPOMI AER\_AI AAI retrieval (Stein Zweers, 2018) uses level 1b earth radiance measurements converted to reflectances using the UVN band solar irradiance measurements, which have a spectral resolution of 0.5 nm (Ludewig et al., 2020). This study uses a selection of 54 TROPOMI orbits between 6 August and 4 September 2019 over the Pacific Ocean, overlaid and averaged based on the equator-crossing time. The orbits have been chosen in such a way that 178 E <  $\eta$  < -140 E,
- where  $\eta$  is the longitude of dayside nadir equator-crossing. We use the TROPOMI offline level 2 AER\_AI aerosol index product from the 340 nm / 380 nm wavelength pair, version 1.2.0. Data is only used if qa\_value > 0.8, retrieved scene albedo at 380 nm is between 0 and 1, and solar zenith angle is smaller than 80°.

#### 2.2 Small-scale effects of clouds on the AAI

- When one zooms-in on a TROPOMI AAI map of a scene with broken clouds, one can always see structures of clouds with high and low AAI values. An example is given in Figure 1, which is a TROPOMI observation over the South Pacific Ocean on 30 August 2018. Please note that no absorbing aerosols are present in this ocean scene, so only clouds can be responsible for the AAI effects.
- The small-scale effects show that clouds have sides with high and low AAI. This is related to the small (nadir) pixel size of TROPOMI ( $3.5 \times 5.5 \text{ km}^2$ ), which is in the order of the size of clouds, and not observed for GOME-2 ( $40 \times 80 \text{ km}^2$  pixels) and OMI ( $13 \times 24 \text{ km}^2$  pixels).

Figure 2 shows the AAI in a cloudy scene over the Pacific Ocean on 30 August 2018 measured by TROPOMI (upper left figure). Large positive AAI values are found next to negative AAI values.

Comparing the AAI map to the true-color image of VIIRS (upper right figure), the TROPOMI top of the atmosphere reflectance (lower left figure) and the TROPOMI calculated scene albedo at 380 nm

Fig. 1. Example of small-scale effects of clouds on the TROPOMI AAI over the southern Pacific Ocean. The large area, about  $6000 \times 5000 \text{ km}^2$  centered at around  $35^\circ \text{ S}$ ,  $170^\circ \text{ W}$ , contains broken clouds, showing positive (red) and negative (blue) AAI values. The size of the zoom-in area is about  $920 \times 760 \text{ km}^2$ . This scene was observed on 30 August 2018.