# Peer review of "Effects of clouds on the UV Absorbing Aerosol Index from TROPOMI"

_Atmospheric Measurement Techniques, 2020_

## Referee Comment (RC1) · Anonymous Referee #1 · 29 May 2020

General comments: This manuscript presents a generally well written study on the effect of clouds on the UV AAI. The presented structural features can be also found on next generation satellite instruments such as Sentinel-4, GEMS, TEMPO. Therefore, I recommend this study for publication in AMT after minor revisions.

Specific comments: Line 24-26: Please provide some references.

Line 41: Please provide some references.

Line 55: I recommend to add one more simple sentence related to physical meaning of using positive AAI for ALH retrieval. For example, only for scenes with a positive AAI value, the ALH retrieval is performed because AAI gets positive for high AOD and absorbing aerosols. For low AOD and scattering aerosols, the TOA reflectance

sensitivity to ALH gets lower.

Line 74: Please add degree symbol here.

Line 76: As I know, there are two products of UVAI in the TROPOMI. One is 354/388, and the another is 340/380. Are there specific reason for choosing 340/380? For the heritage of GOME?

Line 77: Please provide this QA value meaning.

Line 77: I think the word 'Scene albedo' is very important and key word in this manuscript. Scene albedo in this study means, 'Lambertian Equivalent Reflectivity (LER)' or 'Rayleigh Corrected Reflectance (RCR)' in general. But the author probably did not use the 'LER' expression and using 'Scene albedo' because there is a Lambertian assumption for LER. I recommend to add clear physical meaning for 'Scene albedo' , for example, as an expression for Rayleigh corrected reflectance in section 3.1 (or at some appropriate location).

Line 81: 'values. An'  $\rightarrow$  'values. An'

Line138: Please add additional explanation about exact position. "intersection of scanline 3600 and ground pixel 0, scanline 600 and ground pixel 0".

Fig4: Could you please change the figure with higher resolution, or make the character bigger?

Line 142: It is not defined. Please describe as scene albedo.

Line 144: Could you please add exact place of scanline and ground pixel for each case? Other readers may not understand this clearly.

Line 159: I recommend to describe the following sentence, otherwise it could mislead the reader. "The effect of aerosol on the backscattered radiation in the near-UV (320-400 nm), where the ozone absorption is weak and does not affect the interaction between the aerosols and the molecular atmosphere." Fig12-14: Have you ever seen these effect as a function of mean cloud fraction? The author has explained BRDF effect through the manuscript (e.g., Line 199, Line 202, Fig.6, Discussion part), and also mentioned at Line 199 that 'much smaller Rayleigh optical thickness, and causes a strong impact of surface BRDF on the TOA reflectance'. But, I guess the relative portion of surface reflectance from TOA reflectance might be still small at 340 nm and 380 nm especially for the clear sky vegetation (land) region, so that we could assume BRF = LER at those two wavelengths (usually over land). Actually this study are investigated over the Pacific area, so if the LCM, SCM model work well for the small cloud fraction region over Pacific ocean, that could be due to bright ocean surface reflectance. In general, the surface reflectance over ocean would be bright compared to the land surface reflectance due to water-leaving radiance (including the effect of chlorophyll, CDOM) at 340, and 380 nm.

Fig 10: Please add (a), (b) on the figure and caption.

Fig 10: Also, there are many Surface Albedo (As) terms in this thesis, so little bit confusing. So In this Fig 10. the x-axis Surface Albedo +0.05(left) or +0.9(right) corresponds to Surface Albedo in left term of equation (6), or equivalent to Surface Albedo in left term of equation (5), right? And, the x-asis Surface Albedo is actually, d(Surface Albedo) or delta(Surface Albedo), right?

---

## Referee Comment (RC2) · Anonymous Referee #2 · 1 Jun 2020

General Comments:

This study investigates the cloud effects on the AAI with three models: the traditional Lambertian Scene Model (LSM), and two cloud models (i.e., Lambertian Cloud Model (LCM) and Scattering Cloud Model (SCM)) of IPA assumption, primarily at large scales by aggregating TROPOMI data over the Pacific Ocean where absorbing aerosol effect is essentially negligible. Sensitivity studies of the cloud height and surface albedo with a series of scenarios are also conducted through RT simulations.

Strength: this paper presents the first systematic investigations of the cloud effects on the AAI using TROPOMI data of unprecedented high spatial footprints in UV by making comparison of the performance from three models (LCM, SCM, and LSM).

[Figure]

Weakness: results are inconclusive due primarily to the instrumental degradation and calibration issue in current L-1b data. It is difficult to evaluate the performance of the three models with measured radiances of a calibration problem. This limit of the study will be repeatedly pointed out in specific comments below, even though it is beyond the scope of this study. As stated in conclusions, extra future works still remain for improving the operational TROPOMI AAI product including the surface effects.

Overall, this paper is well written and provides useful information of the cloud effects on the AAI for aerosol community. It is appropriate for publication with minor revisions.

Specific Comments:

1. Page 1, lines 18 -19: I have not seen any comparison result of the performance of AAI in terms of footprints sizes (e.g., fine TROPOMI vs. coarse Suomi NPP/OMPS) in this paper. The authors need to discuss such topics in discussion or future plan before stating in abstract.

2. Page 3. Line 77: describe "qa_value" scheme how to derive this quantity.

3. Page 4, Figures 1 and 2: provide regional information of the maps (both longitudes and latitudes ranges) and TROPOMI orbital number.

4. Page 5, lines 120 -125: mostly negative AAI values imply that instrumental effects (not only time dependent degradation but also absolute calibration at 340 and 380nm) appear to be far larger than cloud effects.

5. Page 10-11. Section 3.2 describes the physical principles and interpretation of the Lambertian surface model-based AAI in terms of BRDF concept. In reality, surface BRDF behavior and its effect on the AAI can be far more complex than a simple interpretation of the diffuse to direct light ratio and difficult to say the signs of AAI. Reconsider the change of the title of section 3.2 since it did not show any real surface BRDF effect on the AAI.

6. Page 21, line 410: sun glint features are due to the Fresnel reflection over the ocean

surface under clear sky condition.

7. Page 21, all plots in Figure 12 show large negative AAI values (except sun glints) and a strong cross-track dependence regardless of regions and models, which is not consistent with the results in Torres et al (2018). As stated in many places of this paper (page 2, lines 33-34; page 5, lines 117-119; page 26, lines 477-479), an optimal AAI retrieval would mean that clouds give a neutral AAI, i.e., close to or equal to zero, especially from the statistics of averaging many orbits in this study. The results here indicate that the instrumental effect appears to be far larger than the improvements to forward modeling with cloud models.

8. Page 23, Figure 14, difficult to read legends and labels. Other figures also need to be improved with increased font size and legends.

9. Page 24, Figure 15 shows mostly negative AAI values due to a calibration problem.

10. Page 26, line 463: never shown any real BRDF results.

11. Page 26, lines 473-474: difficult to conclude it because of the instrumental degradation and calibration issue.

12. Page 28, lines 528-536: I disagree. This is not a "subjective" or "preferential" choice issue but a scientific issue. The current TROPOMI AAI product should be further investigated and improved with more effective physical models for absorbing aerosol studies by minimizing other effects such as clouds, surface, and instrument.

13. Page 28, line 540. Other sensors (e.g., Suomi NPP/OMPS-NM AAI and DSCOVR-EPIC AAI) are also capable of detecting such huge smoke plumes at large scales. Clarify an unprecedented "sensitivity" of TROPOMI.

---

## Author Comment (AC1) · 7 Aug 2020

**Response to Reviewers' comments**

"Effects of clouds on the UV Absorbing Aerosol Index from TROPOMI" by Maurits L. Kooreman et al.

**Reviewer #1**

We thank the reviewer for his/her careful reading and for the comments and suggestions, which have improved the manuscript.

Below we give in *blue italic* the reviewer's comment, in black our response, and in purple the changed text in the manuscript.

General comments: This manuscript presents a generally well written study on the effect of clouds on the UV AAI. The presented structural features can be also found on next generation satellite instruments such as Sentinel-4, GEMS, TEMPO. Therefore, I recommend this study for publication in AMT after minor revisions.

We thank the reviewer for the support for this study. The relevance for the three next generation geostationary UV-VIS spectrometers (GEMS being already in orbit) is now added to the outlook in the Conclusions section:

The results of this study are relevant for the future UV-Vis spectrometers with high spatial resolution, like Sentinel-5 on Metop-Second Generation and the three next generation geostationary UV-VIS spectrometers GEMS, TEMPO, and Sentinel-4, all of which will have an AAI product (Kim et al., 2020).

**Specific comments:**

**1. Line 24-26: Please provide some references.**

We have added textbook and review paper references on aerosols and their radiative effects.

**2. Line 41: Please provide some references.**

We have added references on the role of surface albedo in aerosol retrievals.

3. Line 55: I recommend to add one more simple sentence related to physical meaning of using positive AAI for ALH retrieval. For example, only for scenes with a positive AAI value, the ALH retrieval is performed because AAI gets positive for high AOD and absorbing aerosols. For low AOD and scattering aerosols, the TOA reflectance sensitivity to ALH gets lower.

Thank you for this suggestion. Modified text:

Only for scenes with a positive AAI value, the ALH retrieval is performed, because the AAI gets positive for increasing amounts of absorbing aerosols, and especially for elevated absorbing aerosols (De Graaf 2005). Negative AAI values are mostly associated with scattering aerosols and clouds.

**4. Line 74: Please add degree symbol here.**

**Added:**

... that  $178^{\circ} E

We expect that the L1b calibration of TROPOMI will be improved soon, as this is needed for use of the AAI for event and trend detection and for retrievals of aerosol optical thickness and aerosol single scattering albedo from TROPOMI UV L1b data.

To clarify the manuscript, we updated the text in Sect. 2.3:

... also due to a radiometric calibration offset and degradation in the TROPOMI irradiance data.

To further clarify the point that an overall shift of the AAI due to L1b calibration does not impact this study, we added the following text to the Discussion in Sect. 5:

The current TROPOMI L1b calibration bias (including degradation) causes a negative shift in the AAI. However this shift is independent of solar and viewing geometry. So the calibration bias reduces the AAI at a global level, but it does not change the differences in AAI. Therefore it does not impact our study of the orbital distribution of the AAI.

**Specific Comments:**

1. Page 1, lines 18 -19: I have not seen any comparison result of the performance of AAI in terms of footprints sizes (e.g., fine TROPOMI vs. coarse Suomi NPP/OMPS) in this paper. The authors need to discuss such topics in discussion or future plan before stating in abstract.

We agree, so we removed this sentence from the abstract. New text:

The BRDF effect presented here is a first step - more research is needed to explain the small scale cloud effects on the AAI.

2. Page 3. Line 77: describe "qa\_value" scheme how to derive this quantity.

The meaning of the qa\_value is now addressed accordingly:

This quality assurance value is a continuous quality descriptor, varying between 0 (no data) and 1 (full quality data). The value is changed based on observation conditions and retrieval flags and users are recommended to at least ignore data with qa\_value < 0.5 (for details see Apituley et al. (2018)).

**3. Page 4, Figures 1 and 2: provide regional information of the maps (both longitudes and latitudes ranges) and TROPOMI orbital number.**

The longitude and latitude of the image centers are already provided in the captions. Additionally we have provided the orbital numbers.

Fig 1: This scene was observed on 30 August 2018 (orbit 4563). Fig 2: [...] of a cloudy scene over the southern Pacific Ocean (57.3° S, 120.4° W) on 30 August 2018 (orbit 4562).

4. Page 5, lines 120 -125: mostly negative AAI values imply that instrumental effects (not only time dependent degradation but also absolute calibration at 340 and 380nm) appear to be far larger than cloud effects.

As shown above, the calibration offset plus degradation in AAI is about -1 to -1.5, depending on the period. The cloud effects in TROPOMI AAI are much larger: (i) as shown in Figs. 1 and 2, the small-scale cloud effects are up to 6 and larger, namely the differences between the red high values and the blue low values; (ii) as shown in Fig. 3, the large-scale effects are up to 3, namely the differences between the high and low values; (iii) in simulated data in Fig. 7 the cloud effects are around 6. Most importantly, however, as mentioned above in the response to the General comments, we consider in this paper the orbital features (differences in the AAI) due to viewing and solar angle variation - an absolute shift of the AAI by a fixed amount does NOT affect these features.

5. Page 10-11. Section 3.2 describes the physical principles and interpretation of the Lambertian surface model-based AAI in terms of BRDF concept. In reality, surface BRDF behavior and its effect on the AAI can be far more complex than a simple interpretation of the diffuse to direct light ratio and difficult to say the signs of AAI. Reconsider the change of the title of section 3.2 since it did not show any real surface BRDF effect on the AAI.

Indeed, Sect. 3.2 is explaining the principle of the effect of anisotropy (which can be described by a BRDF) on the AAI which is based on a Lambertian surface. We think that this principle is very important, also for complex BRDF functions. Although the figure may be simple, the fact that anisotropy of surfaces and clouds affects the AAI in the UV has many implications.

We changed the title of Sect. 3.2 to: Effect of anisotropy on the AAI

We added a new figure showing the BRDF of Mie scattering clouds for the three optical thicknesses that were used in Fig. 7 for the AAI simulation: COT=1, 8 and 32. We hope this clarifies the relation between BRDF and AAI, as a detailed illustration of Fig. 6 with a realistic BRDF.

New figure: Fig. 7a.

6. Page 21, line 410: sun glint features are due to the Fresnel reflection over the ocean surface under clear sky condition.

We added this explanation to Sect. 3.2, where we discuss sunglint: "in the sunglint, ...due to Fresnel reflection at the ocean surface under clear sky condition."

We further added text on the effect of the sea surface in Sects. 4 and 5, in response to Reviewer #1 comment 13.

7. Page 21, all plots in Figure 12 show large negative AAI values (except sun glints) and a strong cross-track dependence regardless of regions and models, which is not consistent with the results in Torres et al (2018). As stated in many places of this paper (page 2, lines 33-34; page 5, lines 117-119; page 26, lines 477-479), an optimal AAI retrieval would mean

that clouds give a neutral AAI, i.e., close to or equal to zero, especially from the statistics of averaging many orbits in this study. The results here indicate that the instrumental effect appears to be far larger than the improvements to forward modeling with cloud models.

As discussed above, the TROPOMI instrumental effect causes an overall downward shift of the curves in Fig. 12, but does not not change the features and the shape of the cross-section curves. We note that there are large differences between our TROPOMI data and the OMI data of Torres et al. (2018):

- OMI has a 16 x larger pixel size than TROPOMI. Small details visible in TROPOMI AAI are washed out in OMI AAI data (as they are in GOME-2 PMD AAI data).
- Similarly the angular resolution of TROPOMI is much smaller than OMI (and GOME-2 PMD).
- The TROPOMI data are only for clouds over ocean; in Torres et al. aerosols and land effects are present.
- The TROPOMI data are for all latitudes; the OMI study has a limited coverage of latitudes.

We note that there is a cross-track dependence in the Mie UVAI data of Torres et al. (2018), visible in their Fig. 12 for large solar and viewing angles over ocean close to Antarctica. This is a BRDF feature present in all satellite AAI data.

Regarding the statements on a neutral AAI for clouds: in Sect. 1 and Sect. 2 we give the traditional view of the AAI: it should give zero for clouds. This is true when averaged over large areas and large angular ranges. The insight of this paper as given to us by the TROPOMI high resolution data is that an optimal AAI retrieval providing neutral AAI in the absence of aerosols is only possible if one knows the exact BRDF of the underlying scene, being land, ocean, or cloud, or a mixture.

**8. Page 23, Figure 14, difficult to read legends and labels. Other figures also need to be improved with increased font size and legends.**

We increased the figure size. Depending on Copernicus markup we will increase font size if required.

9. Page 24, Figure 15 shows mostly negative AAI values due to a calibration problem.

We agree. Improved TROPOMI L1b calibration will remove the negative bias in the AAI and will shift the entire histograms in the positive direction - but it will not change the width.

**10. Page 26, line 463: never shown any real BRDF results.**

Thank you for the suggestion to show BRDF results; see also point 5. A new figure, Fig. 7a, showing the BRDF of clouds has now been added to Sect. 3.2.

11. Page 26, lines 473-474: difficult to conclude it because of the instrumental degradation and calibration issue.

We do not agree. See our response to the General comments.

12. Page 28, lines 528-536: I disagree. This is not a "subjective" or "preferential" choice issue but a scientific issue. The current TROPOMI AAI product should be further investigated and improved with more effective physical models for absorbing aerosol studies by minimizing other effects such as clouds, surface, and instrument.

The choice of the appropriate physical model for particular use of the AAI is of course a topic of scientific debate. We changed the word "subjective" to "matter of choice".

13. Page 28, line 540. Other sensors (e.g., Suomi NPP/OMPS-NM AAI and DSCOVREPIC AAI) are also capable of detecting such huge smoke plumes at large scales. Clarify an unprecedented "sensitivity" of TROPOMI.

We agree that other satellite instruments can also detect these plumes. We changed the last sentence into:

"... can be detected by TROPOMI with unprecedented sensitivity to small details at high spatial resolution."

---

## Author Comment (AC2) · 7 Aug 2020

Dear reviewer,

Thank you for your elaborate comments. Please find attached the response, provided as supplement to our reply.

We have made a new figure 7, which is also attached.

Best regards, Maurits Kooreman

Please also note the supplement to this comment: https://amt.copernicus.org/preprints/amt-2020-112/amt-2020-112-AC2-supplement.pdf

[Figure]

**Fig. 1.**